# On the Viability of Monocular Depth Pre-training for Semantic Segmentation

## Abstract

We explore how pre-training a model to infer depth from a single image compares to pre-training the model for a semantic task, e.g. ImageNet classification, for the purpose of downstream transfer to semantic segmentation. The question of whether pre-training on geometric tasks is viable for downstream transfer to semantic tasks is important for two reasons, one practical and the other scientific. In practice, if it were viable, one could reduce pre-training cost and bias due to human annotation at scale. If, however, it were not, then that would affirm human annotation as an inductive vehicle so powerful to justify the annotation effort. Yet the bootstrapping question would still be unanswered: How did the ability to assign labels to semantically coherent regions emerge? If pre-training on a geometric task was sufficient to prime a notion of "object", leveraging the regularities of the environment (what Gibson called "detached objects"), that would reduce the gap to semantic inference as a matter of aligning labels, which could be done with few examples. To test these hypotheses, we have designed multiple controlled experiments that require minimal fine-tuning, using common benchmarks such as KITTI, Cityscapes, and NYU-V2: We explore different forms of supervision for depth estimation, training pipelines, and data resolutions for semantic fine-tuning. We find that depth pre-training exceeds performance relative to ImageNet pre-training on average by 5.8% mIoU and 5.2% pixel accuracy. Surprisingly, we find that optical flow estimation, which is a closely related task to depth estimation as it optimizes the same photometric reprojection error, is considerably less effective.

## 1 Introduction

We probe the viability of pre-training on a geometric task, specifically inferring the depth of each pixel, for downstream adaptation to semantic segmentation, which is to assign a label to each group of pixels that belong to the same "object." The baseline for comparison is the widespread practice of supervised pre-training on semantic image tasks such as ImageNet multi-class classification. Training deep neural networks (DNNs) for semantic segmentation in a supervised fashion requires costly pixel-level annotation. Most methods rely on backbones (representation) pre-trained on a different task. While it may seem that semantic segmentation would best be primed by pre-training on semantic tasks, such as image classification, we hypothesize that a seemingly different task, which is geometric, may provide a richer basis from which to fine-tune. The interest in this question arises from the relative ease of acquiring depth data directly, even for specialized domains of interest, through video, multi-view stereo, or range sensor. Taskonomy Zamir et al. (2018), a framework for measuring relationships between visual tasks, suggests that depth estimation is "far" from semantic segmentation, whereas image classification is close. Yet prior work Jiang et al. (2018); Hoyer et al. (2021a); Saha et al. (2021) has shown improved performance when incorporating depth. Of course in the practice one ought to use all the sources of data available, including supervised classification and depth. We test the contrast between the two to evaluate their relative merit and to explore the boostrapping hypothesis that geometric inference necessary for interaction may prime the development of semantic inference.

Another more subtle reason for exploring depth pretraining is that semantic supervision is often performed on heavily human-based datasets, where the photographer who framed a picture meant to convey a particular concept (say, a cup), and therefore took care to make sure that the manifestation of the concept (the image) prominently featured the object by choice of vantage point, illumination,

and lack of occlusion. This bias is mitigated if data is not purposefully organized into "shots." Unfortunately, existing datasets are mostly composed of purposefully framed shots which could obfuscate the analysis. We note that depth can be inferred without any semantic interpretation Julesz (1971) so it can performed equally well for purposefully or randomly captured data. When pre-trained directly in the domain of interest, both depth and semantic segmentation share the same data distribution, thus data selection bias is reduced. We also note that natural scenes exhibit piecewise smooth depth maps, with discontinuities often corresponding to occluding boundaries for detached objects, unlike illumination boundaries or material transitions within the same object that are manifest in the image.

**Methods.** We formalize the main hypothesis in Sect. 3. Since it cannot be tested analytically without knowledge of the joint distribution of test images and labels, we propose an empirical testing protocol. We test on monocular depth models trained under multiple forms of supervision, including structure-from-motion, binocular stereo, and depth sensors. We then change the prediction head of the resulting network, either the final layer or the whole decoder, and fine-tune it for semantic segmentation. We consider depth estimation as a *viable pre-training option if it yields comparable improvements to downstream semantic segmentation task as the current common practice*, i.e. ImageNet pre-training. To this end, we design a series of controlled experiments to test the effect of choice of initialization (Tab. 1, Fig. 2) training with various datasets sizes (Fig. 3), choice of network component to be frozen and fine-tuned (Fig. 4), effect of resolution of training images (Fig. 5), for gauging the feasibility of depth as pre-training. Conclusions are supported by quantitative and qualitative (both in network output and neural activations, i.e., Fig. 6) results. Models are pre-trained on KITTI Geiger et al. (2012), Cityscapes Cordts et al. (2016), and NYU-V2 Nathan Silberman & Fergus (2012), and are evaluated on held-out datasets for semantic segmentation.

**Findings.** Pre-training for depth estimation improves the performance of downstream semantic segmentation across different experimental settings. Particularly, we show that depth estimation is indeed a viable pretraining option as compared to existing methods (Tab. 3), especially given the low cost of obtaining data suitable for depth training (Appendix D.3). For example, compared to classification, using depth on average improves by 5.8% mIoU and 5.2% pixel accuracy. To test the limits of small fine-tuning datasets, we run experiments on 128 fully annotated images down to only 8. Again, monocular depth consistently outperforms ImageNet classification as pre-training (Fig. 3).

Of course, since annotated datasets are available, it makes sense to use them if the goal is to maximize downstream performance. For instance, training a depth network after ImageNet initialization yields better depth estimates than one pre-trained from scratch. This is not surprising considering the volume of annotated images available in classification benchmarks. As a sanity check, we test both a depth network pre-trained from scratch and one trained after ImageNet initialization, and both outperform classification-based pre-training in downstream semantic segmentation. To control the effect of our choice of architecture, we used our pre-trained encoder to initialize a standard semantic segmentation network Chen et al. (2017a). We observed similar findings on Cityscapes and NYU-V2 regardless of how depth training is supervised.

Inferring depth without explicit supervision typically involves minimizing the prediction error, just like optical flow. The difference is the variable with respect to which the minimization is performed: Whereas depth is a positive-valued scalar map, optical flow is a vector field, where each displacement has two degrees of freedom. Somewhat surprisingly, not only does pre-training for depth outperform optical flow, but the latter is often worse than random initialization (Fig. 7).

## 2 RELATED WORK

Pre-training aims to learn a representation (function) of the *test* data that is maximally informative (sufficient), while providing some kind of complexity advantage. In our case, we measure complexity by the validation error after fine-tuning on limited amount of labeled data, which measures the inductive value of pre-training. The recent literature comprises a large variety of "self-supervised" methods that are purportedly task-agnostic. In reality, the task is specified indirectly by the choice of hand-designed nuisance transformations that leave the outcome of inference unchanged. Such transformations are sampled through data augmentation while the image identity holds constant (Contrastive Learning) Chen et al. (2020b;a), or reconstruction Caron et al. (2019). Group transformations organize the dataset into orbits, which contrastive learning tries to collapse onto its quotient, which is a maximal invariant. Such a maximal invariant is transferable to all and only tasks *for which*

Figure 1: **Diagram for different pre-training and fine-tuning setups.** *(a) Common practice: pre-train encoder on ImageNet, attach a decoder (e.g DeepLab), fine-tune the network for semantic segmentation. (b) Our best practice: pre-train the network by monocular depth, fine-tune for semantic segmentation. (c) Cross architecture: for fair comparison with common practice, we pre-train by depth, replace the decoder by DeepLab and fine-tune. (d) To test the quality of pre-trained encoders, we fix the encoders and fine-tune decoders only.*

*the chosen transformation is uninformative*. For group transformation, the maximal invariant can, in theory, be computed in closed form Sundaramoorthi et al. (2009). In practice, contrastive learning are extended to non-group transformations, e.g. occlusions, as seen in language Brown et al. (2020) and images Chen et al. (2019). All self-supervised methods boil down to hand-designed and quantized subsets of planar domain diffeomorphisms (discrete rotations, translations, scaling, reflections, etc.), range homeomorphisms (contrast, colormap transformations) and occlusion masks.

In our case, rather than hand-designing the nuisance transformations assumed to be shared among pre-training and fine-tuning tasks, we let the scene itself provide the needed supervision: images portend the same scene, either from the same timestamp (stereo) or adjacent in the temporal domain (video frames), so their variability defines the union of nuisance factors. These include domain deformations due to ego- and scene motion, range transformations due to changes in illumination, and occlusions. In addition to sharing nuisance variability, pre-training and fine-tuning tasks should ideally also share the hypothesis space. It may seem odd to choose a geometric task, where the hypothesis space is depth, to pre-train for a semantic task, where the hypothesis space is a discrete set of labels. However, due to the statistics of range images Huang et al. (2000) and their similarity to the statistics of natural images Huang & Mumford (1999), this is actually quite natural: A range map is a piecewise smooth function defined on the image domain, whereas a segmentation map is a piecewise constant function where the levels are mapped to arbitrary labels. As a result, the decoder for depth estimation can be easily modified for semantic segmentation. A discussion of this choice, specifically on the representational power of deterministic predictors, in Sect. 5. Jiang et al. (2018); Hoyer et al. (2021a) also utilize depth for semantic segmentation. Jiang et al. (2018) proposes pre-training on relative depth prediction, and Hoyer et al. (2021a;b) utilizes self-supervised depth estimation on video sequences. Our longitudinal experimental results validate their findings. We further validate whether features purely obtained from monocular depth improve semantic segmentation. An extended discussion of these methods is in Appendix D.2. While measuring task relationship or distance Zamir et al. (2018); Ramirez et al. (2019) is related to our work, it is out of our scope; this study is intended to gauge the feasibility of depth as a pre-training option.

Monocular depth methods may use different supervision, either through additional sensors Wong & Soatto (2021), or synthetic data Wong et al. (2021); Lopez-Rodriguez et al. (2020), but none require human annotation. Some use learned regularizers with sparse seeds Wong et al. (2020), or adopt pre-trainings Ranftl et al. (2021). We design experiments agnostic to how depth models are trained.

## 3 FORMALIZATATION OF PRE-TRAINING FOR SEMANTIC SEGMENTATION

Let $x : D \subset \mathbb{R}^2 \to \{0, \ldots, 255\}^3$ be an image, where the domain $D$ is quantized into a lattice, $z : D \to \{1, \ldots, Z\}$ a depth map with $Z$ depth or disparity levels, and $y : D \to \{1, \ldots, K\}$ a semantic segmentation map. In coordinates, each pixel in the lattice, $(i, j) \in \{1, \ldots, N\} \times \{1, \ldots, M\}$ is mapped to RGB intensities by $x(i, j)$, a depth by $z(i, j)$, and a label by $y(i, j)$. Despite the discrete nature of the data and the hypothesis space, we relax them to the continuum by considering the vectors $\mathbf{x} \in \mathbb{R}^{NM3}$, $\mathbf{y} \in \mathbb{R}^{NMK}$ and $\mathbf{z} \in \mathbb{R}^{NM}$. With a slight abuse of notation, $y \in \{1, \ldots, K\}$ denotes a single label and $\bar{y} \in \mathbb{R}^K$ its embedding, often restricted to a one-hot encoding.

Now, consider a dataset $\mathcal{D}_z = \{\mathbf{x}_t^i, \mathbf{z}_t^i\}_{i,t=1}^{V,T_i}$ comprised of $V$ video sequences each of length $T_i$ and synchronized depth maps measured by a range sensor – typical datasets supporting depth estimation

| Pre-training | Fine-tune All | | | | | | Freeze Encoder | | | |
| | ResNet18 | | ResNet50 | | DeepLabV3$^{\dagger}$ | | ResNet18 | | ResNet50 | |
| | mIoU | P.Acc | mIoU | P.Acc | mIoU | P.Acc | mIoU | P.Acc | mIoU | P.Acc |
|---|---|---|---|---|---|---|---|---|---|---|
| None | 41.35 | 70.75 | 44.66 | 73.37 | 21.93 | 52.32 | 41.24 | 70.52 | 37.72 | 67.38 |
| ImageNet | 45.15 | 72.39 | 44.65 | 73.06 | **43.39** | **72.66** | **33.33** | 65.34 | 32.03 | 62.53 |
| Depth-Rand | 46.00 | 72.43 | 49.90 | 76.28 | 43.43 | 71.34 | 43.02 | 72.38 | 45.79 | **74.71** |
| Depth | **50.20** | **76.39** | **50.92** | **77.34** | **43.77** | 72.68 | **46.53** | **74.42** | **46.55** | 74.48 |

Table 1: **Semantic segmentation accuracy on KITTI.** *Unupervised depth as pre-training improves semantic segmentation accuracy under all settings. Our best practice (in* blue*) improves common practice (in* red*) by 7.53% mIoU and 4.68% pixel accuracy.* Green*: Freezing the encoder with ImageNet pre-training is worse than no pre-training (random initialization). DeepLabV3*$^{\dagger}$*: with ResNet50 encoder.*

may include just video sequences, or both modalities. In the case of video, pre-training for depth estimation yields a map $\phi_w : \mathbf{x} \mapsto \mathbf{z}$, parametrized by weights $w$, via

$$w = \arg\min_{w, g_t} \sum_{i,j,n,t} \ell(x_{t+1}^n(i,j), \hat{x}_t^n(i,j)) \tag{1}$$

where $\hat{x}_t$ is the warping of an image $x$ from $t$ to $t+1$ based on camera motion $g_t$ (see Appendix B.1). If depth measurements $z_t$ are available, then $w$ minimizes a loss on the estimated depth $\hat{z}_t$ via

$$w = \arg\min_{w} \sum_{i,j,n,t} \ell(z_{t+1}^n(i,j), \hat{z}_t^n(i,j)) \tag{2}$$

The goal is to use these representations, or embeddings, as encodings of the data to then learn a semantic segmentation map. In practice, the representations above are implemented by deep neural networks, that can be truncated at intermediate layers thus providing embedding spaces larger than the respective hypothesis spaces. We refer to the parts before and after this intermediate layer *encoder* and *decoder*, respectively. We overload the notation and refer to the encoding as $\mathbf{h} = \phi_w(\mathbf{x})$ for both depth estimation and other pre-training methods, presumably with weights $w'$, assuming they have the same encoder architecture. The goal of semantic segmentation is then to learn a parametrized map $\psi_{w''} : \mathbf{h} \mapsto \mathbf{y}$ using a small but fully supervised dataset $\mathcal{D}_s = \{\mathbf{x}^n, \mathbf{y}^n\}_{n=1}^N$, by minimizing some loss function or (pseudo-)distance in the hypothesis space $d(\mathbf{y}, \hat{\mathbf{y}})$, where

$$w'' = \arg\min_{w} \sum_{n=1}^N d(\mathbf{y}^n, \psi_w(\mathbf{h}^n)) \tag{3}$$

plus customary regularizers. In the aggregate, we have a Markov chain: $\mathbf{x} \longrightarrow \mathbf{h} = \phi_w(\mathbf{x}) \longrightarrow \mathbf{y} = \psi_{w''}(\mathbf{h}) = \psi_{w''} \circ \phi_w(\mathbf{x})$ for depth estimation, and $\mathbf{x} \longrightarrow \hat{\mathbf{h}} = \phi_{w'}(\mathbf{x}) \longrightarrow \mathbf{y} = \psi_{w''} \circ \phi_{w'}(\mathbf{x})$ for other pre-training methods. A representation obtained through a Markov chain is optimal (minimal sufficient) only if the intermediate variable $\mathbf{h}$ or $\hat{\mathbf{h}}$ reduces the Information Bottleneck Tishby et al. (2000) to zero. In general, there is information loss, so we **formalize the key question** at the center of this paper as whether the two Information Bottleneck Lagrangians satisfy the following:

$$H(\mathbf{y}|\mathbf{h}) + \beta I(\mathbf{h};\mathbf{x}) \overset{?}{\leq} H(\mathbf{y}|\hat{\mathbf{h}}) + \beta' I(\hat{\mathbf{h}};\mathbf{x}) \tag{4}$$

where $\beta$ and $\beta'$ are hyperparameters that can be optimized as part of the training process, and $I, H$ denotes the (Shannon) Mutual Information and cross-entropy respectively. If the above is satisfied, then pre-training for depth estimation is a viable option, or even better than pre-training with another method. It would be ideal if this question could be settled analytically. Unfortunately, this is not possible, but the formalization above suggests a protocol to settle it empirically.

To test this empirically, we consider the validation error on a supervised dataset $\mathcal{D}_s$ as a proxy for residual information (Appendix A). We conduct fine-tuning under several configurations: with respect to $w''$ using $\mathcal{D}_s$, i.e. yielding a comparison of the raw pre-trained back-bones (encoders, $w, w'$), or with respect to *both $w''$ and $w$* (for depth estimation) or $w'$ (for other pre-training methods). Finally, all four resulting models can be compared with one obtained by training from scratch by optimizing a generic architecture with respect to $w''$ alone. These settings are visualized in Fig. 1.

## 4 EXPERIMENTS

In this section (and Appendix C), we present experiments under 51 different settings with varying architectures and pre-trainings. Unless specified, each experiment is repeated 4 times and we report

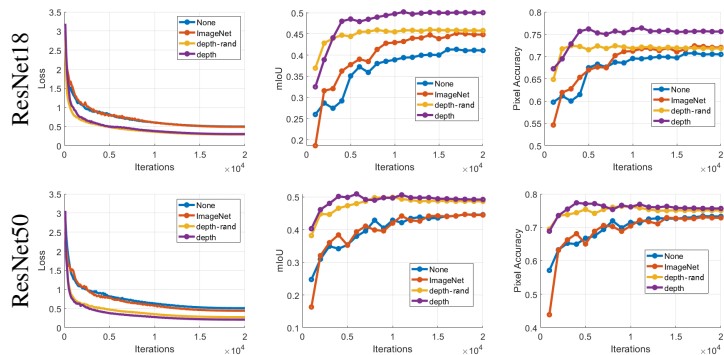

Figure 2: **Comparison between different network initializations.** *Models initialized by depth pre-training (unsupervised) train faster and achieve higher final accuracy.*

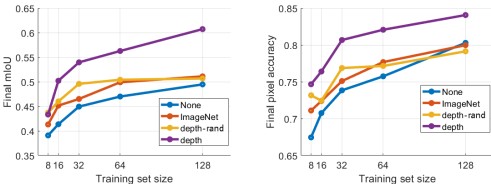

Figure 3: **Final accuracy vs different training set size.** *Under all training set sizes, our best practice constantly outperforms ImageNet pre-trained. Encoder: ResNet18.*

the average. Due to the enormous amount of experiments, we adopt the KITTI-semantic dataset for ablation and sensitivity studies for its size, which allows for extensive controlled experiments. Then we extend the experiments to Cityscapes and NYU-V2. The three datasets also represent three different modes for learning monocular depth: Minimizing prediction error in videos i.e. unsupervised (KITTI), supervising by binocular stereo (Cityscapes), and using time-of-flight (TOF) depth sensor for groud-truth (NYU-V2). Each type of depth supervision can easily be obtained, and is often available (i.e. video) in many existing datasets, making it feasible to pre-train for depth on the same dataset as segmentation. This feasibility of acquiring in-domain depth data is an advantage of using depth as pre-training. A detailed discussion is in Appendix D.3).

## 4.1 KITTI DATASET

KITTI contains 93000 images for depth training with 200 densely annotated images for semantic segmentation. Segmentation results are evaluated by mean IoU (mIoU) and pixel-level accuracy (P.Acc). We randomly choose a small training set (i.e. few-shot) of 16 images and limit data augmentation to horizontal flips to highlight the impact of pre-training, except for Fig. 3 where we test on different training set partitions. We use Monodepth2 Godard et al. (2019) for depth pre-training. For semantic segmentation, we replace the last layer of the decoder with a fully connected layer, using the finest scale of the multi-scale output. We use a resolution of $640\times192$ following Monodepth2, and also provide results at higher resolution and evaluate consistency across resolutions (also see Appendix C.1). We use ResNet18 and ResNet50 encoders due to their compatibility with various network architectures and the wide availability of pre-trained models. Fig. 1 summarizes our experimental setup and Tab. 1 summarizes the outcomes. In all cases, depth pre-training improves segmentation accuracy. Our best practice improves over the current common practice of ImageNet pre-training with DeepLab V3 head, by 7.53% mIou and 4.68% pixel accuracy.

**Full model.** Fig. 2 shows the evolution of training loss and model accuracy. Depth pre-training outperforms ImageNet and random initialization. ImageNet pre-training slightly improves over random initialization on ResNet18, but shows almost identical performance to random initialization on ResNet50. Depth also speeds up training, taking $\sim 5000$ iterations to converge, while ImageNet takes 15000 to 20000. Similar results are also observed for full-resolution images (Appendix C.1).

**Different training set size.** Fig. 3 shows that depth pre-training improves final segmentation scores over all dataset partitions. When training samples increase (e.g 128), ImageNet and depth pre-training (from random initialization) are comparable to random initialization, but depth pre-training initialized with ImageNet yields the best results.

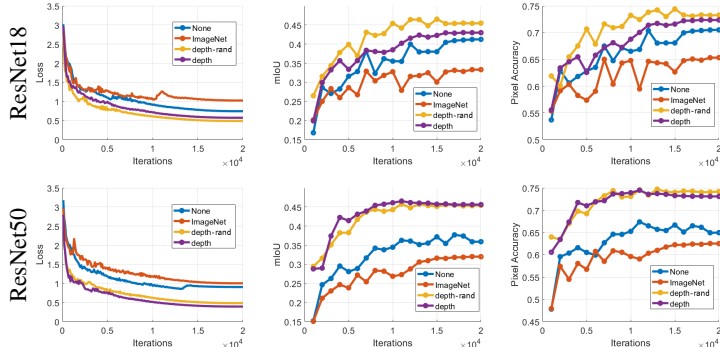

Figure 4: **Frozen encoder results.** *Using an encoder pre-trained by depth significantly outperforms one with random weights and one for ImageNet classification. Note that in this experiment, ImageNet pre-training performs worse than random initialization.*

|  | Pre-training | mIoU | P.Acc |
|---|---|---|---|
| **Res. 18** | None | 41.35 | 70.75 |
| | ImageNet | 45.15 | 72.39 |
| | Depth (encoder only) | **46.69** | **75.04** |
| **Res. 50** | None | 44.66 | 73.37 |
| | ImageNet | 44.65 | 73.06 |
| | Depth (encoder only) | **46.99** | **73.57** |
| **ViT-L** | ImageNet | 57.53 | 81.48 |
| | Depth (encoder only) | **58.12** | **81.94** |

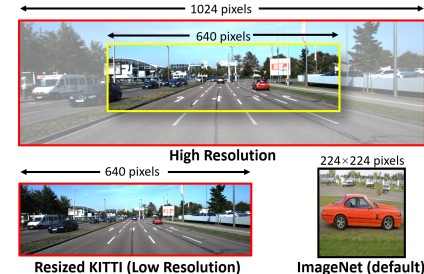

Table 2: **Initializing with depth encoder and random decoder.** *Initializing with just the depth encoder and a random decoder outperforms ImageNet weights, but is worse than initializing with both encoder and decoder from the depth network (see Tab. 1). The findings also hold for ViTs.*

Figure 5: **Mismatch between object scales in ImageNet and KITTI.** *ImageNet models are trained with a fixed input size and objects of interest tend to have a similar scale. However, objects in semantic segmentation dataset vary drastically in scales. Pre-training for depth provides robustness to scale change.*

**Frozen encoder.** We freeze pre-trained encoders, and fine-tune the decoder only, testing the ability of features from pre-trained encoders to capture semantics. With both ResNet18 and ResNet50, encoder pre-trained for depth significantly outperforms random initialization and ImageNet pre-trained (Fig. 4). It is surprising that ImageNet pre-training is detrimental in this case (after a grid search over learning rates): worse than fixed random weights. This suggests that while classification is a semantic task, it removes semantic information about the *scene* due to the object-centric bias in datasets. ImageNet pre-training tends to favor generic texture features, that do not capture object shape, which makes fine-tuning the decoder difficult for learning segmentation. We conjecture that these uncontrolled biases in ImageNet and the task gap between classification and semantic segmentation cause difficulties in directly predicting segmentation without fine-tuning the encoder.

**Initializing with pre-trained encoder only.** To eliminate the effect of the depth-initialized decoder and only test the encoder, we replace the decoder by 'fresh' randomly initialized weights and fine-tune the whole network. Tab. 4.1 (marked blue) shows that depth pre-training outperforms ImageNet when the effect of pre-training is isolate to the encoder. This is also supported by the neural activations in Fig. 6 where, unlike ImageNet pre-training, the regions activated by the encoder after depth pre-training align well with semantic boundaries. Nonetheless, the decoder does play a role in segmentation accuracy: Initializing the whole network with depth pre-training still performs the best – an advantage that is not afforded by a classification head.

**Results on Visual Transformers.** Given that Visual Transformers (ViTs) Dosovitskiy et al. (2020) necessitate extensive pre-training, conducting comprehensive ablation studies on ViTs, as feasible with ResNet, becomes impractical due to computational constraints. Nonetheless, we conduct experiments using the DPT Ranftl et al. (2021) architecture, and report the best results in Tab. 4.1 after an exhaustive search for the optimal learning rate. Our findings remain consistent. Notably, we identified the optimal learning rate for this experiment to be 5e-8, with larger learning rates yielding suboptimal results. This suggests that pre-trained ViTs inherently provide robust representations, requiring minimal fine-tuning for semantic segmentation in comparison to CNNs like ResNet.

**Cross architecture.** To test whether depth pre-training favor our particular network architecture, we use the same encoders to initialize DeepLab V3 and follow the same training procedure as common

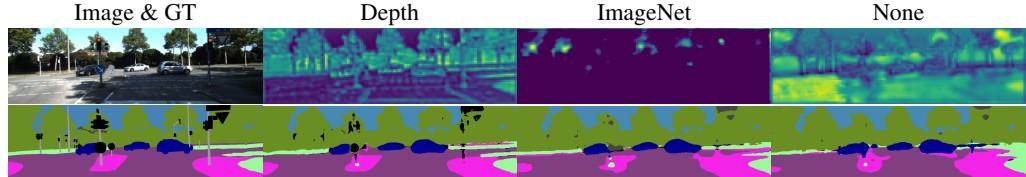

Figure 6: **Neural activation and semantic segmentation result.** *We visualize the neural activation map for a shallow layer of ResNet 18 encoder trained from different initializations, and their corresponding segmentation results. Boundaries are better aligned to semantic boundaries in our model. Better viewed zoomed-in.*

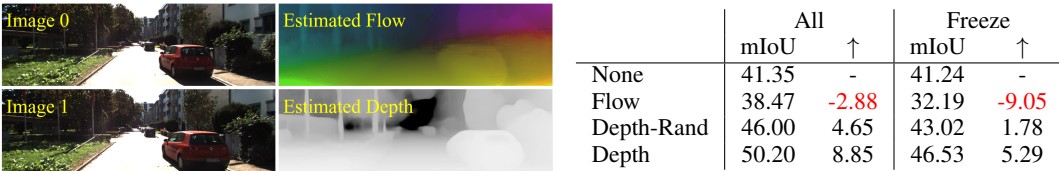

| | All | | Freeze | |
|---|---|---|---|---|
| | mIoU | ↑ | mIoU | ↑ |
| None | 41.35 | - | 41.24 | - |
| Flow | 38.47 | -2.88 | 32.19 | -9.05 |
| Depth-Rand | 46.00 | 4.65 | 43.02 | 1.78 |
| Depth | 50.20 | 8.85 | 46.53 | 5.29 |

Figure 7: **Depth helps, flow hurts.** *Both pre-trained by minimizing photometric reconstruction error, monocular depth outperforms optical flow. This stems from the fact that inferring depth from a single image is ill-posed so the network learns rich semantic priors over the structures within a scene. In contrast, any discriminative features will support the correspondence search, so the flow network is not constrained to learning semantics.*

practice, i.e., ImageNet initialization. All pre-trainings significantly improve accuracy compared with random initialization (Tab. 1). Depth model trained from scratch provides a same level of performance as ImageNet pre-training, while depth trained after ImageNet improves over ImageNet.

**Robustness to object scales.** Given a fixed resolution, 'primary' objects in object-centric datasets tend to have a similar scale. For example, ImageNet models are trained on 224×224, thus cars typically have a size of 100 to 200 pixels. In KITTI, however, cars appear at different scales, varying from a few to a few thousand pixels. Fig. 5 illustrates the scale mismatch between datasets. On the other hand, depth pre-training can be done in the same domain as segmentation, hence has robustness to object scales. We examine such robustness: Pre-training on one resolution, fine-tuning on another. Since higher resolution images contains smaller scales (yellow box in Fig. 5), pre-training on them should still work on smaller images. Pre-trained on smaller resolution, however, should work poorly on larger images. Our results validate this conjecture: The former achieves a final mIoU of 48.54 (ImageNet: 45.15) while the latter diverges during training in multiple independent trials.

**Neural Activation.** We visualize the activations of the ResNet18 encoder by Grad-CAM Selvaraju et al. (2017). Grad-CAM is originally designed for classification; we modify it for segmentation by inspecting the gradient of neural response with respect to the summation of predicted labels for pixels instead of one single class label. We visualize shallow layers (before pooling) for high spatial resolution. Fig. 6 shows neural activation maps and segmentation outputs from depth pre-training align with semantic boundaries. This confirms not just the similarities between natural and range image statistics discussed in Huang & Mumford (1999), but also the bias introduced by classification, as activations of ImageNet pre-trained encoder do not resemble object boundaries.

**Comparison with optical flow and other pre-training methods.** One hypothesis for the effectiveness of depth pre-training is that the process leverages the statistics of natural scenes where simply-connected components of the range map often correspond to semantically consistent regions. Thus, fine-tuning simply aligns the range of two piece-wise smooth functions. We challenge this hypothesis by trying optical flow, which also exhibits a piecewise-smooth range and is obtained by minimizing the same photometric error. We train optical flow on a siamese network with two shared-weight encoders. While both optical flow and depth capture multiply-connected object boundaries (Fig. 7), using encoders pre-trained for optical flow is detrimental. We conjecture that optical flow does not capture the stable inductive bias afforded by the static component of the underlying scene. Specifically, optical flow is compatible with an underlying 3D geometry only when the scene is rigid, but rigidity is not enforced when inferring optical flow. In contrast, depth estimation forces recognition of rigidity and discards moving objects as outliers (which then enables isolating them, also beneficial to semantic segmentation). Experiments on ResNet50 draw the same conclusion (Tab. 3).

In Tab. 3, for completeness, we report results on supervised pre-training by semantic segmentation on MS-COCO Lin et al. (2014). Unsurprisingly, pre-training on the same task with additional an-

| Pre-training | mIoU | P.Acc | Pre-training | mIoU | P.Acc |
|---|---|---|---|---|---|
| Supervised Segmentation | **51.28** | 74.88 | Contrastive (DINO) | 44.19 | 71.36 |
| Depth | 50.92 | **77.34** | Optical Flow | 42.72 | 71.80 |
| Reconstruction (MAE) | 47.18 | 74.16 | Contrastive (MOCO V2) | 37.04 | 65.91 |

Table 3: **Comparison with different pre-trainings.** *Encoder: ResNet50; Reconstruction: by inpainting randomly corrupted regions (masked autoencoding); Supervised Segmentation: trained on MS-COCO.*

| | Full | | | | Controlled | | | |
|---|---|---|---|---|---|---|---|---|
| | Training | | Validation | | Training | | Validation | |
| | mIoU | P.Acc. | mIoU | P.Acc. | mIoU | P.Acc. | mIoU | P.Acc. |
| None | 76.10 | 95.41 | 63.97 | 93.76 | 73.82 | 95.26 | 60.43 | 93.07 |
| ImageNet | 81.84 | 96.46 | 70.41 | 95.75 | 76.70 | 95.71 | 61.80 | 93.40 |
| Depth | 83.46 | 96.80 | **73.17** | **95.24** | 77.42 | 95.82 | 62.57 | 93.46 |
| Depth-cropped | **86.80** | **97.41** | 72.22 | 95.01 | **79.90** | **96.24** | **65.09** | **94.00** |

Table 4: **Segmentation accuracy on Cityscapes.** *Similar to KITTI, pre-training for depth improves segmentation accuracy. Interestingly, when pre-training on cropped $256 \times 256$ patches, the model achieves higher performance under the controlled settings (with limited data augmentation and fewer epochs).*

notated data yields good performance. Depth estimation, despite not needing additional annotation, yields on-par performance. We also report results on masked autoencoding. We remove random rectangular regions from images, and the network aims to reconstruct the original image. Also known as 'inpainting' Bertalmio et al. (2000), it is considered an effective method for feature learning Pathak et al. (2016). It yields inferior performance compared with depth. We conjecture that this is due to artificial rectangular masking which does not respect the natural statistics, while monocular depth estimation yields occluded regions that border on objects' silhouettes. We also provide results initializing the encoder with contrastive learning (MOCO V2 Chen et al. (2020b) and DINO Caron et al. (2021)). While they also remove bias in human annotation for classification, similar to supervised classification, they are still prone to the inherent inductive bias in pre-training data.

## 4.2 CITYSCAPES DATASET

Cityscapes contains 2975 training and 500 validation images. Each image has a resolution of $2048 \times 1024$ densely labeled into 19 semantic classes. The dataset also has 20000 unlabeled stereo pairs with a disparity map, converted to depth via focal length and camera baseline. Like KITTI, Cityscapes is also an outdoor driving dataset. Here we minimize an L1 loss between depth estimates and depth computed from stereo. We modify the prediction head of DeepLabV3 to train for depth, then re-initialize the last layer of the decoder for semantic segmentation. Due to limitations in compute, we are unable to reproduce the original numbers. For a fair comparison, we retrained Chen et al. (2017b) using the discussed pre-training methods and finetuned under the same training protocol i.e. batch size, augmentations, schedule, etc. (details in Appendix B.4). Tab. 4 summarizes the outcomes. We present not only the most favorable results from an extensive training process, but also results from a more controlled approach involving restricted data augmentation and fewer training iterations. Remarkably, our observations remain consistent across both scenarios: Pre-training with monocular depth improves over ImageNet initialization and random initialization.

One may argue that since depth and semantic segmentation maps are both piece-wise smooth, adapting from depth is naturally easy if the model is aware of each pixel's relative position in the image. In order to test this statement, instead of pre-training for depth on the full image, we train depth on randomly cropped $256 \times 256$ patches, and the model has no spatial awareness of the position of a patch in the image, so depth is purely estimated by local information. This practice (Depth-cropped) surprisingly improves semantic segmentation results under controlled settings, showing that using depth as pre-training goes beyond a simple mapping from one smooth function to another. Interestingly, this approach significantly improves training accuracy in the full setting but leads to a slight reduction in validation accuracy, suggesting a potential issue of overfitting. Future research is necessary to delve into this intriguing phenomenon. Another noteworthy observation is that when the model is pre-trained using depth data, it exhibits superior training performance when initialized with a higher learning rate of 0.1, as opposed to the conventional 0.01 used for ImageNet initialization. Conversely, employing an initial learning rate of 0.1 when the model is initialized with ImageNet weights can be detrimental and may result in divergence. These findings suggest that pre-training the network with depth estimation may lead to a smoother local loss landscape.

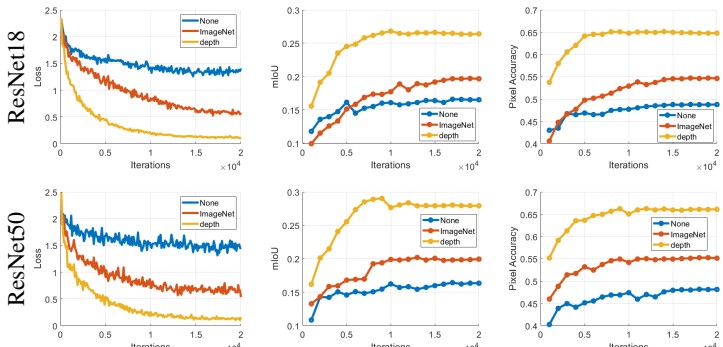

Figure 8: **Results on NYU-V2.** *Similar to KITTI and Cityscape, initializing the model with depth pre-trained weights trains faster and significantly improves semantic segmentation accuracy.*

### 4.3 NYU-V2 DATASET

Unlike KITTI and Cityscapes, NYU-V2 is an indoor dataset that contains 795 densely annotated images for training and 654 for testing. There are also 407024 unannotated frames available, with synchronized depth images captured by a Microsoft Kinect. Since our claims are agnostic to how depth estimation models are trained, we pre-trained for depth using supervision from a depth sensor, which are now common in most platforms, including mobile phones.

Unlike outdoor driving, which commonly feature sky on top, and road and vehicles in the middle of the image with largely planar camera motion, indoor scenes are characterized by more complex layouts with 6 DoF camera motion – yielding images that are even less likely to resemble the object-centric ones commonly observed in classification datasets. This may be why initializing the model with depth pre-trained weights significantly improves semantic segmentation accuracy with both ResNet18 and ResNet50 encoders (see Fig. 8). Note that pre-training by depth yields faster convergence, which is also our observed in KITTI.

### 5 DISCUSSION

Depth estimation requires multiple images *of the same scene* and does not require human-induces priors and biase Julesz (1964), unlike image classification that relies entirely on induction: We can associate a label to an image because that image has *something* in common with *some other* image, portraying a different scene, that some annotator attached a particular label to. That inductive chain has to go through the head of human annotators, which are biased in ways that cannot be easily quantified and controlled. Depth estimation from binocular or motion imagery does not require any induction and can be performed *ab-ovo* at inference time.

Our findings are agnostic to how depth is attributed to a single image: One can perform pre-training using monocular videos, stereo, structure light, LIDAR, or even human annotation. The important point is that human annotation is not needed. Of course, if millions of annotated images are available, we want to be able to use that information. One of our pre-training minimizes the photometric reprojection error, used by many predictive and generative approaches. However, unstructured displacement field is in general not compatible with a rigid motion. Only if this displacement field has the structure of an epipolar transformation Sundaramoorthi et al. (2009) is the prediction task forced to encode the 3D structure of the scene. This may explain why video prediction is not as effective for pre-training despite many attempts Jin et al. (2020); Wu et al. (2020); Wang et al. (2020).

One limitation of depth is that it requires the use of a calibrated camera, so one cannot use generic videos harvested from the web. There is nothing in principle preventing us from using uncalibrated cameras, simply by adding the calibration matrix $K$ to the nuisance variables. While the necessary conditions for full Euclidean reconstruction are rarely satisfied in consumer videos (for instance, they require cyclo rotation around the optical axis, not just panning and tilting), the degrees of freedom that cannot be reconstructed are moot as they do not affect the reprojection. Future extensions to uncalibrated cameras have the potential to unlock a virtually unlimited volume of training data.

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

## A   FORMALIZATION OF EMPIRICAL SET-UP

The Information Bottleneck Lagrangian in Sect. 3 cannot be computed because we do not have access to the analytical form of the joint distributions of the variables $(\mathbf{x}, \mathbf{h})$, $(\mathbf{h}, \mathbf{y})$ and $(\mathbf{x}, \hat{\mathbf{h}})$, $(\hat{\mathbf{h}}, \mathbf{y})$. Even defining, let alone computing, the Shannon Information for random variables that are deterministic maps $\phi(\cdot)$ of variables $\mathbf{h}$ defined in the continuum is non-trivial Achille et al. (2019). It is possible, however, to bound the Information Bottleneck, which is not computable, with the *Information Lagrangian* Achille & Soatto (2018), which only depends on the datasets $\mathcal{D}_y, \mathcal{D}_z$ and $\mathcal{D}_s$. However, the bound depends on constants that are functions of the complexity of the datasets, which are different for different tasks, which would render them useless in answering the question in equation 4. Therefore, we consider the validation error as a proxy of residual information:

$$L_z(w''|w) = \sum_{\mathbf{h}^n = \phi_w(\mathbf{x}^n)} - \log p_{w''}(\mathbf{y}^n|\mathbf{h}^n) \simeq H(\mathbf{y}|\mathbf{h}) \tag{5}$$

for pre-training using depth estimation, and

$$L_y(w''|w') = \sum_{\hat{\mathbf{h}}^n = \phi_{w'}(\mathbf{x}^n)} - \log p_{w''}(\mathbf{y}^n|\hat{\mathbf{h}}^n) \simeq H(\mathbf{y}|\hat{\mathbf{h}}) \tag{6}$$

for pre-training using another pre-training method. The complexity terms $I(\mathbf{h}; \mathbf{x})$ and $I(\hat{\mathbf{h}}; \mathbf{x})$ are minimized implicitly by the capacity control mechanisms in the architecture (*i.e.,* the maps $\phi_w(\cdot|\mathcal{D}_z)$ and $\phi_{w'}(\cdot|\mathcal{D}_y)$), for instance pooling; in the regularizers, for instance weight decay and data augmentation Achille et al. (2019); and in the optimization, for instance stochastic gradient descent Chaudhari & Soatto (2018). The losses above are computed by summing over the samples in the validation set $\mathcal{D}_s = \{\mathbf{x}^n, \mathbf{y}^n\}$.

## B   IMPLEMENTATION DETAILS

### B.1   WARPING

$$\hat{x}(i, j) = x \circ \pi_{g,z}^{-1}(i, j) \tag{7}$$

is the *warping* of an image $x$ onto the image plane of another camera related to it by a change of pose $g \in SE(3)$, through the depth map $z$, via a *reprojection map* $\pi^{-1}$

$$\pi^{-1}(i, j) = K_+\pi(R_t K^{-1}[i, j, 1]^T z(i, j) + T_t) \tag{8}$$

that embeds a pixel $(i, j)$ in homogeneous coordinates, places it in the camera reference frame via a calibration matrix $K$, back-projects it onto the scene by multiplying it by the depth $z(i, j) = \phi_w(x(i, j)|\mathcal{D}_z)$ and then transforming it to the reference frame of another camera with a rigid motion $g = (R, T)$, where the rotation matrix $R \in SO(3)$ and the translation vector $T \in \mathbb{R}^3$ transform the coordinates of spatial points $P \in \mathbb{R}^3$ via $P \mapsto RP + T$. Here $\pi$ is a canonical perspective projection $\pi(P) = [P(1),\ P(2)]/P(3)$ and the calibration map $K_+$ incorporates quantization into the lattice. Here, we assume that the intrinsic calibration matrix $K$ is known, otherwise it can be included among nuisance variables in the optimization along with the inter-frame pose $g_t$ when minimizing the reprojection error $\ell$ in equation 7.

Note that the reprojection error could be minimized with respect to $w$, which is shared among all images and yields a depth map through $z_t = \phi_w(x_t|\mathcal{D}_z)$, or directly with respect to $z_t$ in equation 8, which does not require any induction. Since the goal of pre-training is to capture the inductive bias we adopt the former and discuss it in detail in Sect. 5.

### B.2   TRAINING AND EVALUATION DETAILS ON KITTI

#### B.2.1   PRE-TRAINING FOR UNSUPERVISED DEPTH ESTIMATION.

Monodepth2 is trained by optimizing a linear combination of photometric reprojection error and an edge-aware local smoothness prior

$$L(w') = w_{ph}\ell_{ph} + w_{sm}\ell_{sm}, \tag{9}$$

where

$$\ell_{ph} = \sum_{i,j,n,t} \left(1 - \text{SSIM}(x_{t+1}^n(i,j), \hat{x}_t^n(i,j))\right) +$$
$$\alpha |x_{t+1}^n(i,j) - \hat{x}_t^n(i,j)|_1 \tag{10}$$

$\hat{x}_t$ is the warped image equation 7 and $\ell_{sm}$ is the edge-aware smoothness prior

$$\ell_{sm} = \sum_{i,j,n,t} |\partial_X z_t^n(i,j)| e^{-|\partial_X x_t^n(i,j)|} +$$
$$|\partial_Y z_t^n(i,j)| e^{-|\partial_Y x_t^n(i,j)|}, \tag{11}$$

$w_{ph}$ and $w_{sm}$ are hyper-parameter weights.

### B.2.2 Semantic segmentation fine-tuning.

Following the notation in Sect. 3 in the main paper, we denote with $\phi_{w''} : \mathbf{x} \mapsto \hat{\mathbf{y}}$ the semantic segmentation network to be fine-tuned, which maps an image $\mathbf{x}$ to a semantic label $\mathbf{y}$. Note that $\phi_{w'}$ (classification network) is parameterized by weights $w'$ of the 'encoder' network, and $\phi_w$ (depth network) is parameterized by weights $w$ of both the 'encoder' and a 'decoder'. When pre-trained for classification, we initialize the encoder part of $w''$ by $w'$ and the decoder part by random weights; when pre-trained for depth, we initialize both encoder and decoder in $w''$ using $w$, except for the last layer, where we change to a randomly initialized fully-connected layer with soft-max. During semantic segmentation fine-tuning, we update $w''$ (see equation (11) and (12) in the main paper) by minimizing the cross entropy loss

$$L(w''|\bullet) = \sum_{i,j,n,k} -\log(\hat{y}^n(i,j)))\mathbb{1}(y^n(i,j) = k) \tag{12}$$

where $i, j$ are the pixel coordinates, $n$ is the number of images in the training set, $k$ is the class label, $\hat{y} = \phi_{w''}(\mathbf{x}^n)$ is the network output. This is implemented via the Negative Log Likelihood (NLL) loss in Pytorch. Under different experimental settings, we either update all parameters in $w''$ or only the decoder part of $w''$ to minimize equation 12.

KITTI contains 21 semantic classes, and we fine-tune on all of them. However, since we do not use a separate dataset with segmentation labels, many of the semantic classes are seldom seen (*e.g.*, 'train', 'motorcycle'), especially with just 16 training images. In such cases, these classes always receive zero IoU, which downweights the mIOU metric (but still yields high P.Acc). Therefore, we compute mIoU on a subset of 7 representative classes unless stated otherwise. The results of 21 classes exhibit the same trends.

### B.2.3 Image normalization.

In fine-tuning, we apply the same image normalization that is consistent with the pre-training step. If the network is pre-trained by ImageNet classification, we normalize the image values by mean=[0.485, 0.456, 0.406], std=[0.229, 0.224, 0.225]; if pre-trained by Monodepth2, we normalize the image values to [0,1]; we also normalize to [0,1] when training from random initialization.

### B.2.4 Optimizer.

After a grid search, we choose $1 \times 10^{-5}$ as the initial learning rate for the ADAM optimizer. The learning rate is updated by a standard cosine learning rate decay schedule in every iteration.

### B.3 Details for training optical flow

To train a neural network $f_\theta$ parameterized by $\theta$ to estimate optical flow for a pair of images $(x_t, x_{t+1})$ from time step $t$ to $t + 1$, we leverage the photometric reconstruction loss by minimizing a color consistency term and a structural consistency term (SSIM) between an image $x_{t+1}$ and its reconstruction $\hat{x}_t$ given by the warping $x_t$ with the estimated flow $f_\theta(x_t, x_{t+1})$:

$$\ell_{ph} = \sum_{n,i,j,t} \lambda_{co}\big(|x_{t+1}^n(i,j) - \hat{x}_t^n(i,j)|\big)+ $$
$$\lambda_{st}\big(1 - SSIM(x_{t+1}^n(i,j), \hat{x}_t^n(i,j))\big), \tag{13}$$

where $\hat{x}_t = x_t \circ f_\theta(x_t, x_{t+1})$, $f_\theta(\cdot) \in \mathbb{R}^{2HW}$ and $\lambda_{co} = 0.15$, $\lambda_{st} = 0.15$ are the weights for color consistency and SSIM terms for $L_{ph}$, respectively.

Additionally, we minimize an edge-aware local smoothness regularizer:

$$\ell_{sm} = \lambda_{sm} \sum_{n,i,j} \lambda_X(i,j)|\partial_X f_\theta(x_t^n, x_{t+1}^n)(i,j)|+ $$
$$\lambda_Y(i,j)|\partial_Y f_\theta(x_t^n, x_{t+1}^n)(i,j)| \tag{14}$$

where $\lambda_{sm} = 5$ is the weight of the smoothness loss, $\partial_X, \partial_Y$ are gradients along the x and y directions, and the loss for each direction is weighted by $\lambda_X := e^{-|\partial_X x_t^n|}$ and $\lambda_Y := e^{-|\partial_Y x_t^n|}$ respectively.

We trained $f_\theta$ using Adam optimizer with $\beta_1 = 0.9$ and $\beta_2 = 0.999$. We set the initial learning rate to be $5 \times 10^{-4}$ for the first 25 epochs and decreased it to $5 \times 10^{-5}$ another 25 epochs. We used a batch size of 8 and resized each image to $640 \times 192$. Because KITTI has two video streams from left and right stereo camera, we randomly sample batches from each stream with a 50% probability. Training takes $\approx 10$ hours for ResNet18 backbone and $\approx 20$ hours for ResNet50.

## B.4 DETAILS FOR TRAINING ON CITYSCAPES

Different from KITTI, we perform pre-training for depth on a modified DeepLabV3 architecture. We change the prediction head to output a single depth channel instead of 19 classification channels. We follow the standard image normalization of ImageNet training. We use ADAM optimizer with an initial learning rate of 1e-4 and linear learning rate decay. Data augmentations include brightness, contrast, saturation and random horizontal flips. Each model is trained on a single Nvidia GeForce GTX 1080 Ti GPU with a batch size of 8 for 15000 iterations, which takes approximately 6.5 hours. During fine-tuning, we re-initialize the prediction head for segmentation. Each model is trained on two Nvidia GeForce GTX 1080 Ti GPUs. We use SGD optimizer with polynomial learning rate decay. We did a learning rate search and report the best performing settings. We find using 0.1 as the initial learning rate yields optimal results when pre-trained by depth, while 0.01 works the best with ImageNet pre-training. All other settings follow the original DeepLabV3 implementation. Limited by the GPU memory we crop training images to $512 \times 512$ patches and set the batch size to be 8. Each model is trained for 60000 iterations (30000 iterations under 'controlled' settings), taking 15 hours, and we report the accuracy after the final epoch.

## B.5 DETAILS FOR TRAINING ON NYU-V2

Because NYU-V2 provide image and depth map pairs, similar to Cityscapes, we directly train Monodepth2 $\phi_w$ by minimizing an $L_1$ loss:

$$\ell_{L_1} = \sum_{n,i,j} \mathbb{1}(z^n(i,j) > 0)\big(|\phi_w(x^n)(i,j) - z^n(i,j)|\big), \tag{15}$$

where $\phi_w(x)$ is the predicted depth for an image $x$ and $z$ is the ground truth depth from a Microsoft Kinect. Because the ground truth is only semi-dense, this loss is only computed where there is valid depth measurements *i.e.,* $z(i,j) > 0$.

We trained Monodepth2 using the ADAM optimizer with $\beta_1 = 0.9$ and $\beta_2 = 0.999$. We set the initial learning rate to be $1 \times 10^{-4}$ for 5 epochs and decreased it to $5 \times 10^{-5}$ another 5 epochs for a total of 10 epochs. We used a batch size of 8 and resized each image to $448 \times 384$. For data augmentation, we perform random brightness, contrast and saturation adjustments within the range of $[0.80, 1.20]$ with a 50% probability. Pre-training the depth network takes $\approx 19$ hours for ResNet18 backbone and $\approx 30$ hours for ResNet50. After pre-training, we do semantic segmentation fine-tuning on the training set. As with KITTI, we apply the ADAM optimizer with $1 \times 10^{-5}$ initial

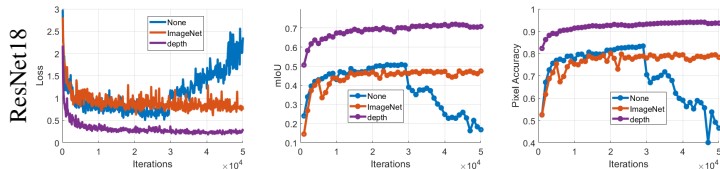

Figure 9: **Training ResNet with high resolution images.** *Similar to low resolution images, pre-training on depth also improves semantic segmentation accuracy on high resolution images.*

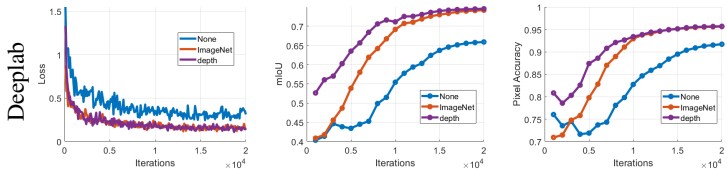

Figure 10: **Training Deeplab on high resolution images.** *Compared to lower resolutions, performance improves for different initializations. Training loss (left) is similar between ImageNet and depth initialization, but mIOU (center) and pixel accuracy (right) are higher for depth initialization for similar loss values.*

learning rate and cosine learning rate decay, and restrict data augmentation to horizontal flipping. Limited by GPU memory, for ResNet18 we use batch size 32, and for ResNet50 we use batch size 16. All models are trained for 20000 iterations. Each experiment is repeated by four independent trials.

## C EXPERIMENTAL RESULTS (CONTINUED)

### C.1 RESULT ON HIGH-RESOLUTION IMAGES (KITTI)

In Fig. 9 we show results trained with full resolution (1024×320) and full dataset (200 images), in which case we train for an extended number of iterations (50000). Pre-training by depth still outperforms ImageNet pre-training and random initialization. Note that in this case using random initialization is unstable and training diverges after 30000 iterations.

We also conduct experiments on high-resolution images. Results are consistent with low-resolution experiments. Although both ImageNet and depth initialization converged to the same level of accuracy, the depth pre-trained model shows a high accuracy in earlier iterations. This is interesting given that training loss of both ImageNet and depth pre-train are almost the same. We conjecture that the features learned by single-image depth estimation are more conducive to segmenting 'bigger' classes (e.g. 'road', 'building') which are mostly rigid and take up larger portions of the image.

To further investigate this behavior, in Figure 11 we plot the mIoU curve on all (21 classes), and a scatter plot of mIoU versus pixel accuracy in the training process. While mIoU on all 21 classes follows the trend observed on 7 classes, the scatter plot shows that at each same level of mIoU, models pre-trained by depth have higher pixel accuracy. This validates our conjecture that the depth-pre-trained model learns 'bigger' classes faster, since higher performance on these classes will result in high pixel accuracy as they have more pixels.

### C.2 TRAINING LOSS ON CITYSCAPES

In Fig. 12 we show training loss of early iterations on Cityscapes shows that training on small crops of the image improves convergence. We revisit the question we posed in Table 4 in the main text regarding this phenomenon below.

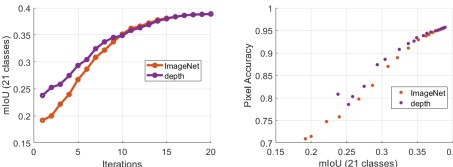

Figure 11: **mIoU v.s. pixel accuracy during training DeepLab.** *At each same level of mIoU, model pre-trained by depth has a higher pixel accuracy. This validates our conjecture that the depth-pre-trained model learns 'bigger' classes faster.*

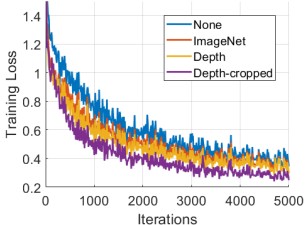

Figure 12: **Training loss of early iterations on Cityscapes.** *In early training iterations, the loss of Depth-cropped decreases significantly faster than other initializations.*

### C.3 VISUALIZATIONS OF SEGMENTATION

Fig. 13 shows head-to-head comparisons of representative outputs from semantic segmentation models that have been pre-trained with ImageNet classification, and pre-trained with monocular depth estimation. As discussed in Sect. 4.1 Neural Activations, pre-training on ImageNet biases the model towards capturing generic textures exhibited in the image rather than that object shape (see Fig. 6). This results in a loss of details when fine-tuning for the downstream segmentation task where the goal is precisely to capture object boundaries. We illustrate the drawback of pre-training on ImageNet in Fig. 13-left where the model over-predicts the street sign in the center (highlighted in yellow) and under-predicts the pedestrians on the right (highlighted in red) with spurious predictions of the vehicle class alongside them. This is in contrast to pre-training with monocular depth estimation, where the downstream segmentation model is able to capture the edge between the street sign and building regions in the middle as well as the small pedestrian regions in the far distance on the right. Additionally, we show in Fig. 13-right that an ImageNet pre-trained model has difficulty outputting the same class for a consistent surface like the sidewalk on the left, which is unlike a model pre-trained on depth. This is also supported by our results in Appendix C and Sect. 4.1 in the main text where the "larger" object classes tend to be more easily learned (higher general P.Acc) by a model pre-trained on depth than one trained on ImageNet.

While classification is a semantic task, training for it requires discarding nuisances including objects (other than one that is front and center, see neural activations in Fig. 6 and results with a frozen pre-trained encoder in Fig. 4) that may exist in the background of an image. Pre-training for depth, on the other hand, involves solving correspondence problems, which naturally requires estimating the boundaries of objects and consistent locally-connected, piecewise smooth surfaces. Thus, one may hypothesize that pre-training to predict depth or geometry process makes it more straightforward to assign these surfaces with a semantic class or label i.e. road, pedestrian, building. We conjecture that this may play a role in sample complexity as illustrated by the performance improvements observed over ImageNet pre-training when training with fewer samples (Fig. 3).

## D EXTENDED DISCUSSION

### D.1 CROPPING SIZE DURING PRE-TRAINING

One may object that once a depth map has been estimated, a semantic map is just a matter of aligning labels. However, depth is not necessarily a piecewise constant function, although it is generally

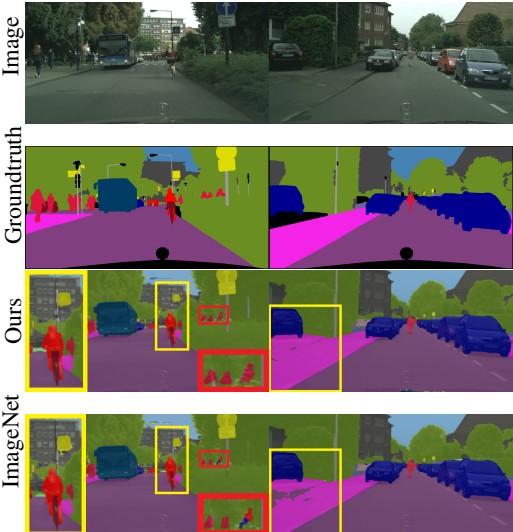

Figure 13: **Head-to-head comparison between ImageNet and monocular depth estimation pre-training for semantic segmentation on Cityscapes.** *We visualize the representative outputs of segmentation models pre-trained on monocular depth (row 3) and ImageNet (row 4). Pre-training with depth enables sharper object boundaries, i.e. street sign highlighted in yellow, pedestrians highlighted in red, and more consistent class predictions on large objects like the sidewalk on the left in the right column. We note that pre-training on ImageNet also yields spurious predictions like the vehicle class next to the pedestrians (highlighted in red) and the street sign class on the street light in the middle of the left image. Better viewed zoomed-in and in color.*

piecewise smooth. So, for instance, the road at the bottom center of KITTI images is a slanted plan that does not correspond well to any constant value, yet the model converts it into a consistent class.

One may also object that slanted planes at the bottom center of the visual field have a strong bias towards being labeled "road" given the data on which the model is trained.

For this reason, we conducted the experiments shown in Tab. 4, whereby we select random crops of 3% of the size of the image, and use those for pre-training rather than the full image. This way, there is no knowledge of the location of the patch of the slanted plane relative to the image frame. We were expecting a degradation in performance, but instead observing an improvement.

While in theory full consideration of the visual field is more informative, provided sufficient training data, due to the limited volume of the training set and the strong biases in the training data, breaking the image into smaller patches and discarding their relation (position on the image plane) may help break the spurious dataset-dependent correlations and lead to better generalization after fine-tuning.

This conclusion is speculative, and we leave full testing to future work. The important aspect of this experiment is to verify that fine-tuning semantic segmentation after depth pre-training is not just a matter of renaming a piecewise smooth depth field into a piecewise constant labeled field.

## D.2 RELATION TO PRIOR ARTS

The primary objective of this study is to conduct an exhaustive examination of the adoption of monocular depth as a pre-training technique for semantic segmentation and to compare it with classification (which has been the de-facto approach for weight initialization). It is acknowledged that prior research Hoyer et al. (2021a;b); Jiang et al. (2018) has also recommended the use of depth as a pre-training method. While we are not claiming originality in using depth, our study stands out as the first to systematically investigate this issue under different setups including network architecture, resolution, and supervision. Nevertheless, there are also distinctions between our study and these methods.

Jiang et al. (2018) performs pre-training by estimating relative depth on images, while we focus on estimating absolute depth either through supervised or unsupervised pre-training. We present results that contradict those of Jiang et al. (2018). While Jiang et al. (2018) claim that ImageNet is

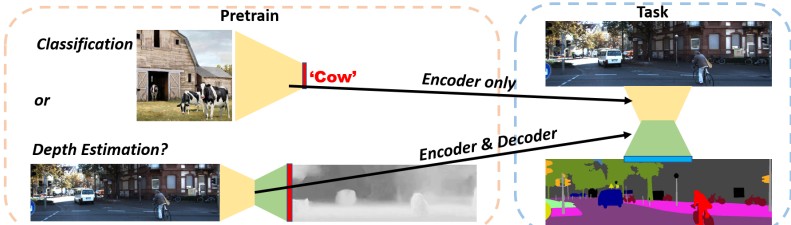

Figure 14: *Image classification introduces uncontrolled biases when used to pre-train semantic segmentation networks, where one requires an additional decoder. Depth estimation, on the other hand, is not subject to semantic bias in the pre-training dataset, and eliminates the need for human annotation, and can easily adapt the pre-training dataset to the domain of interest. The question is whether such a process can improve performance and reduce dependency of annotated pre-training datasets in fine-tuning for semantic segmentation.*

the better pre-training approach, our findings show better performance when using our depth pre-training as compared to ImageNet initialization alone. The difference in findings is suprising as we observe consistent performance boost across our experiments; one possible reason may be that camera calibration is available to us (also a valid assumption for real-world scenarios that require semantic segmentation, such as self-driving), and it is well-known that un-calibrated depth estimation is a more difficult problem than the calibrated case. Another difference (and the possible reason for the contradicting results) is that Jiang et al. (2018) synthesizes relative depth from optical flow estimation network, and train a depth prediction network by minimizing L1 difference between predictions and relative depth. The authors also state that potential errors in optical flow will impact (and compound the error in) the downstream depth.

Hoyer et al. (2021a;b) use self-supervised depth prediction as a proxy task for semantic segmentation, where the features are regularized to stay close to ImageNet features. However, our experiments demonstrate that in some cases, ImageNet features may not support the task of semantic segmentation as observed in Fig. 6, and also empirically validated in Fig. 13. Even when we train the depth model with ImageNet initialization, we do not assume that the features should remain largely unchanged. Nonetheless, we are not against using ImageNet, as it is off-the-shelf and can be useful to expedite training depth estimation networks, likely due to the formation of generic filter banks in the early layers.

We would emphasize that our paper validates the findings in prior works Jiang et al. (2018); Hoyer et al. (2021a;b), yet provides a more comprehensive longitudinal study (across different architectures, datasets, supervision, etc.) towards using monocular depth as pre-training for semantic segmentation.

### D.3 INSIGHTS ON THE INTUITIONS AND FEASIBILITY OF USING MONOCULAR DEPTH AS PRE-TRAINING

ImageNet classification is widely regarded as the primary task for pre-training in the context of semantic segmentation. The dataset comprises more than 14 million annotated images, collected through crowd-sourcing with the assistance of around 15,000 paid annotators. Empirical studies have consistently confirmed the advantages of using ImageNet for initial model training, resulting in substantial enhancements in the performance of semantic segmentation tasks. This outcome is unsurprising, as both image classification and semantic segmentation involve understanding the meaning of objects in images. It's important to note that obtaining detailed pixel-level annotations for semantic segmentation is a costly and resource-intensive process. Consequently, datasets specifically tailored for semantic segmentation are considerably smaller in scale compared to ImageNet, often differing by several orders of magnitude in terms of data size.

In contrast, the KITTI dataset, which is widely recognized for its use for depth estimation for driving scenarios, consists of approximately 86,000 training images. These images are captured at a rate of 10 frames per second, resulting in less than three hours of driving video. Collecting this type of driving video data is relatively straightforward and requires only a driver and a dashboard camera. In addition to continuous video data, collecting training data for monocular depth estimation can also utilize hardware, such as multi-view stereo systems or depth sensors like Time-of-Flight (ToF) and Lidar. What is common among these data sources is that they demand minimal labor and resources

compared to the extensive efforts needed for ImageNet data collection. This affordability allows for the scaling up of initial model training directly within the domain of interest.

From an intuitive perspective, it is more natural to transfer knowledge between tasks that share semantic similarities, such as between different semantic tasks, than to transfer between tasks with distinct characteristics, like transitioning from a geometric task to a semantic task. The main problem is that image classification is *defined* by induction and therefore does not only entail, but *requires* a strong inductive bias, which opens the door to potentially pernicious side-effects. Induction is required because, continuing the example of the image labeled as "cow", there is nothing in the image of a scene that would enable one to infer the three-letter word "cow." The only reason we can do so is because the present image resembles, in some way implicitly defined by the training process, *different images, of different scenes,* that some human has tagged with the word "cow." However, since images, no matter how many, are infinitely simpler than even a single scene, say the present one, this process is not forced to learn that the extant world even exists. Rather, it learns regularities in images, each of a different scene since current models are trained with data aiming to be as independent as possible, agnostic of the underlying scene.

Now, one may object that monocular depth estimation is itself undecidable. This is why monocular depth estimators are trained with either multiple views (motion or stereo), or with some form of supervision or partial backing from additional modalities, such as sparse depth from lidar Kuznietsov et al. (2017) or other range sensor Smisek et al. (2013). Then, a depth estimate is just a statistic of the learned prior. As a result, depth estimation networks should never produce *one* depth estimate for each image, but rather a distribution over depth maps, conditioned on the given image Yang & Soatto (2018); Yang et al. (2019). Given an image, every depth map is possible, but they are not equally likely. The posterior over depth given an image then acts as a prior in depth inference using another modality, *e.g.,* unsupervised depth completion. The mode of this conditional prior can then be used as a depth estimate if one so desires, but with the proviso that whatever confidence one may place in that point estimate comes from inductive biases that cannot be validated

One additional objection is that, since depth requires optimization to be inferred, and the choice of the loss function is a form of transductive bias, that is no less arbitrary than inductive bias. But this is fundamentally not the case, for the optimization residual in transductive inference refers to the data *here and now*, and not to data of different images in different scenes. In other words, the optimization residual is informative of the confidence of our estimate, unlike the discriminant from an inductively-trained classifier Der Kiureghian & Ditlevsen (2009).

In some cases, one enriches (augments) the predictive loss with manually engineered transformation, calling the result "self-supervised learning." Engineered transformations include small planar group transformations like translation, rotation, scaling, reflections, and range transformations such as contrast transformation or colorization. But for such group transformations, learning is not necessary since we know the general form of the maximal invariant with respect to the entire (infinite-dimensional) diffeomorphic closure of image transformations Sundaramoorthi et al. (2009). One exception is occlusion and scale changes, which are not groups once one introduced domain quantization. So, pre-training using masking, ubiquitous in language models, does learn a preudo-invariant to occlusion, but rather than simulating occlusions, one can simply observe them in the data (a video). The only supervision is then one bit, provided temporal continuity: The fact that temporally adjacent images portray the same scene.

In this paper, we aim to bypass the artificial constraints imposed both by supervised classification, and self-supervision. Instead, we simply use videos to pre-train a model for depth estimation, without supervision. Then, one can use supervision to fine-tune the model for semantic segmentation. These two tasks are seemingly antipodal, yet depth estimation outperforms image classification when used as pre-training for semantic segmentation.

This also addresses one last objection that one can move to our thesis, which is that, since ImageNet data is available, it makes sense to use it. What we argue here is that, actually, it does not. The argument is corroborated by evidence: Pre-training on a geometric task improves fine-tuning a semantic task, even when compared with pre-training with a different semantic task. Nonetheless, ImageNet pre-training can be useful to expedite training depth estimation networks, likely due to the formation of generic filter banks in the early layers.

