# OpenReview forum: "On the Viability of Monocular Depth Pre-training for Semantic Segmentation"
_ICLR.cc/2024/Conference — Submitted to ICLR 2024_

### Official Review · Reviewer_N1Lo · 2023-11-01

**Soundness:** 3 good
**Presentation:** 1 poor
**Contribution:** 2 fair
**Rating:** 3
**Confidence:** 3

**Summary:**

This paper is about using monocular depth estimation training as pre-training for the semantic segmentation task. The idea is quite simple, whether such a pre-training can be better than pretraining with classification task on ImageNet. Extensive experiments are carried out to verify the hypothesis. The conclusion of this paper is that compared to classification, using depth estimation as pretraining on average improves the segmentation performance.

**Strengths:**

+ The idea to use monocular depth estimation as pretraining for semantic segmentation is sensible, considering that it is relatively easy to collect depth data.
+ The experimental results support the hypothesis.

**Weaknesses:**

- It would be interesting to see the segmentation performance on PASCAL VOC and COCO with the depth pretraining.
- The writing is poor, sometimes the notations are not well defined or clarified. For instance, in EQ.1, since we are doing pretraining for depth estimation, why in the loss function, the depth term is missing?  Instead, there is the wrapped image? Similarly, the input of the loss function in EQ.2 is not clarified as well.
-  Some results are not sufficiently analyzed. For example, in the left figure of Fig.3, it seems that with larger training size, the bigger gain could be obtained.  However, this is counter-intuition, which needs proper explanations.

**Questions:**

We see the the main comparisons are between ImageNet classification-based pre-training vs Depth pre-training. One thing to consider is that depth pre-training happens in-domain with the same data to be fine-tuned. My question is whether the superiority of depth pre-training over ImageNet pre-training mainly relies on in-domain knowledge or data distribution. What if the depth pre-training is conducted on a different dataset rather than the one to be fine-tuned. Will it still has such significant improvement over classification pre-training?

---

> ### Author Response · Authors · 2023-11-23
> **Author response to the initial review**
>
> **W1:** *It would be interesting to see the segmentation performance on PASCAL VOC and COCO with the depth pre-training.*
>
> **R:** We cannot report results on PascalVOC and COCO due to a lack of camera calibration on both datasets, unlike KITTI, CityScapes, and NYU-V2. Consequently, we cannot perform depth pre-training accordingly. Although it is possible to use a more generic depth model (e.g., DPT trained on a mixed dataset) for fine-tuning on both datasets, we attempted to conduct the experiment as suggested by the reviewer but encountered restrictions in compute resources (batch size restricted by GPU memory).
>
> **W2:** *The writing is poor, sometimes the notations are not well defined or clarified. For instance, in EQ.1, since we are doing pre-training for depth estimation, why in the loss function, the depth term is missing? Instead, there is the wrapped image? Similarly, the input of the loss function in EQ.2 is not clarified as well.*
>
> **R:** Eq. 1 and 2 refer to the standard structure-from-motion formulation, which is a convention for inferring depth from multi-view/video data (please see Appendix B.1 for a discussion regarding this). Inferring depth does not require depth measurements as supervision. One only needs images with sufficiently exciting textures that are co-visible between them. We will clarify this in the paper revision.
>
> **W3:** *Some results are not sufficiently analyzed. For example, in the left figure of Fig.3, it seems that with larger training size, the bigger gain could be obtained. However, this is counter-intuition, which needs proper explanations.*
>
> **R:** For Fig. 3-left, a larger training set size for fine-tuning semantic segmentation improves results. This shows that pre-training with depth not only improves sample complexity but also scales well with respect to it, which is not the case for pre-training with ImageNet, where it saturates after 64 samples.
>
> **Q1:** *We see the the main comparisons are between ImageNet classification-based pre-training vs Depth pre-training. One thing to consider is that depth pre-training happens in-domain with the same data to be fine-tuned. My question is whether the superiority of depth pre-training over ImageNet pre-training mainly relies on in-domain knowledge or data distribution. What if the depth pre-training is conducted on a different dataset rather than the one to be fine-tuned. Will it still has such significant improvement over classification pre-training?*
>
> **R:** The superiority of depth pretraining over ImageNet is not contingent on in-domain pretraining. The last row of Table 2 shows initializing with weights from DPT, which does not include KITTI data for pretraining. Despite being out-of-domain, depth pretraining still improves over ImageNet pretraining. Please also see *'A general note to the reviewers and AC'* for a detailed discussion.
>
> We hope this response addresses the reviewer's concerns. If there are further concerns, we are more than willing to provide additional explanations.

---

### Official Review · Reviewer_Ynow · 2023-11-01

**Soundness:** 3 good
**Presentation:** 3 good
**Contribution:** 2 fair
**Rating:** 5
**Confidence:** 3

**Summary:**

The paper shows that depth pre-training exceeds performance relative to ImageNet pre-training on various downstreaming tasks (segmentation / depth prediction). The authors conduct extensive experiments with various architectures and settings.

**Strengths:**

1. The paper conducts extensive experiments to demonstrate that pre-training on depth images is beneficial to downstream tasks (depth / segmentation).

2. It’s interesting to see depth pre-training helps for some downstreaming tasks in a few-shot manner.

3. The paper is in general easy to understand.

**Weaknesses:**

1. The paper studies how depth-pretraining helps for downstreaming tasks. Since the main downstreaming task is still depth prediction, I feel that pre-training on depth prediction shows some improvements that are not that impressive to the community. Though segmentation is also studied on CityScape, maybe testing on more datasets would be more convincing.

2. Regarding optical flow, I wonder if there is more clarification on why optical flow is not a good choice for pre-training? Is it because the prediction is more inaccurate / harder? Furthermore, how about optical flow as a downstreaming task? Does depth pre-training still help?

3. In Figure 2, I wonder why Depth-rand is on par with ImageNet pre-trained for ResNet-18, but the observation is very different for ResNet-50. I wonder if the authors have any thoughts on this?

4. Suppose more images are available, would the pre-training in depth still be beneficial? For example, in Figure 3, what if there are hundreds of images, maybe using the full training set? Similarly, for segmentation, how much gain could the model obtain if there are more images? Is it also the case that only 16 images are used for fine-tuning for the results obtained?

5. It would be helpful to clarify the experiment settings in Table 1 at the beginning of the experiment section. I wonder if the authors may further clarify the difference between  ‘Depth-Rand’ and ‘Depth’?

6. Since this is mainly a paper comparing different existing strategies, in general it is of little technical novelty.

**Questions:**

Please see my questions above. In general, I feel the paper shows some interesting findings, but I am not sure if there is enough contribution to the community.

---

> ### Author Response · Authors · 2023-11-21
> **Author response to the initial review**
>
> **W1:** *The paper studies how depth-pretraining helps for downstreaming tasks. Since the main downstreaming task is still depth prediction, I feel that pre-training on depth prediction shows some improvements that are not that impressive to the community. Though segmentation is also studied on CityScape, maybe testing on more datasets would be more convincing.*
>
> **R:** There may be some misunderstanding but the downstream task in this manuscript is always semantic segmentation. We show that pre-training on monocular depth estimation can in general improve downstream semantic segmentation.
>
> **W2:** *Regarding optical flow, I wonder if there is more clarification on why optical flow is not a good choice for pre-training? Is it because the prediction is more inaccurate / harder? Furthermore, how about optical flow as a downstreaming task? Does depth pre-training still help?*
>
> **R:** We provided a detailed discussion about using optical-flow on page 7. We validated that the flow obtained by pre-training is accurate (also see Figure 7 for visualization), so the improvement is not related to the inaccuracy or difficulty of the task. Moreover, see our response to Reviewer gFUe, where using miscalibrated camera intrinsic in depth estimation (so that the results are not accurate) can also lead to improvement.
>
> Using depth pre-training may help optical flow in specific domains. However current main-stream flow methods (e.g. RAFT, PWC-Net) do not incorporate a standard backbone, so a direct comparison is infeasible. We would like to remind the reviewer that the purpose of this paper is to validate whether training on a geometry task can actually help improve a seemingly distantly related semantic task.
>
> **W3:** *In Figure 2, I wonder why Depth-rand is on par with ImageNet pre-trained for ResNet-18, but the observation is very different for ResNet-50. I wonder if the authors have any thoughts on this?*
>
> **R:** One possibility is ResNet50 has bigger capacity than ResNet18, so there can be more local minimum and the model is subject to overfitting. We will include the discussion in the paper as suggested.
>
> **W4:** *Suppose more images are available, would the pre-training in depth still be beneficial? For example, in Figure 3, what if there are hundreds of images, maybe using the full training set? Similarly, for segmentation, how much gain could the model obtain if there are more images? Is it also the case that only 16 images are used for fine-tuning for the results obtained?*
>
> **R:** On KITTI, we keep the 16-image training set except for Figure 3, as stated on page 5. We do this “few-shot” experiment to highlight the effect of pre-training. However, on Cityscapes, we do use the full training set, and adopting monocular depth as pre-training still yields improvement.
>
> **W5:** *It would be helpful to clarify the experiment settings in Table 1 at the beginning of the experiment section. I wonder if the authors may further clarify the difference between ‘Depth-Rand’ and ‘Depth’?*
>
> **R:** We will clarify. Depth-rand refers to depth model trained from scratch (randomly initialized weights), while depth refers to the best depth model, trained from standard ImageNet initialization. We show that despite not using any annotation, only training for monocular depth on 3 hours of driving video, can leads to similar results to training on ImageNet, which is collected through crowd-sourcing with the assistance of around 15,000 paid annotators. Nevertheless, since ImageNet is there, one can make best use of it to train a better depth model, which further improving the performance yielding the best result.
>
> **W6:** *Since this is mainly a paper comparing different existing strategies, in general it is of little technical novelty.*
>
> **R:** Regarding the innovation of the paper, please refer to *A general note to the reviewers and AC*.
>
> We hope this response addresses the reviewer's concerns. If there are further concerns, we are more than willing to provide additional explanations.

---

### Official Review · Reviewer_gFUe · 2023-11-02

**Soundness:** 3 good
**Presentation:** 3 good
**Contribution:** 2 fair
**Rating:** 6
**Confidence:** 2

**Summary:**

In this paper, the authors explore how pre-training a model to infer depth from a single image compares to pre-training the model for a semantic task for the purpose of downstream transfer to semantic segmentation. The intuition of their work is to avoid human annotation by pre-training a model on the depth estimation task in which the Ground Truth can be acquired through video, multi-view stereo, or range sensor. They carefully design experiments to prove that depth pre-training exceeds performance relative to ImageNet pre-training, and optical flow estimation is less effective.

**Strengths:**

1. This paper is well-organized, which help readers easy to read and understand. Expecially, the intuition of this paper is described clearly and sounds make sense. The topic studied in this paper is critical to the industry. It is a good practice guidline.
2. The experiments on multiple datasets are extensive, which covers almost all potential variations. It helps the authors conclude multiple guidances and helps convince readers. The conclusions are usefual to industry.
3. More details are reported in the appendix. These information is a good addition to the main paper. Implementation details, training details, more results analysis are clear and should be able to help other researchers to duplicate their work.

**Weaknesses:**

The whole paper sounds like a experimental report. My biggest concern about this paper is its lack of innovation. Many existing work has explored ways to combine depth and segmentation, while this paper did more extensive experiments to summarize a more helpful pipeline. Beside that, there is no other contribution.

**Questions:**

1. Almost all experiments in this paper assume the depth ground truth are reliable and calibrated with the RGB images. What if the depth  ground truths are not reliable (have noises, does not calibrated with RGB images)? How does it impact the performance of the proposed pipeline?

---

> ### Author Response · Authors · 2023-11-21
> **Author response to the initial review**
>
> Thank you for your support of the paper.
>
> **W1:** *The whole paper sounds like a experimental report. My biggest concern about this paper is its lack of innovation. Many existing work has explored ways to combine depth and segmentation, while this paper did more extensive experiments to summarize a more helpful pipeline. Beside that, there is no other contribution.*
>
> **R:** Regarding the innovation of the paper, please refer to *A general note to the reviewers and AC*.
>
>
> **Q1:** *Almost all experiments in this paper assume the depth ground truth are reliable and calibrated with the RGB images. What if the depth ground truths are not reliable (have noises, does not calibrated with RGB images)? How does it impact the performance of the proposed pipeline?*
>
> **R:** This question is very interesting. As suggested by the reviewer, we introduce miscalibration to the depth model by allowing camera intrinsics to vary by up to 10%. Surprisingly, we achieve on-par improvement compared with using the ground-truth calibration on CityScapes. We posit that the connection between depth and object semantics is influenced by the object scale. A miscalibrated camera affects the object's scale, but it does not significantly change the object's semantics. We are more than happy to include this discussion in the paper.
>
> We hope this response addresses the reviewer's concerns. If there are further concerns, we are more than willing to provide additional explanations.

---

> > ### Comment · Reviewer_gFUe · 2023-11-22
> >
> > Thank you authors' responses. All of my questions are answered. No more questions. Thanks.

---

### Official Review · Reviewer_r9ta · 2023-11-07

**Soundness:** 3 good
**Presentation:** 2 fair
**Contribution:** 2 fair
**Rating:** 5
**Confidence:** 2

**Summary:**

This paper explores leveraging monocular depth estimation as a pre-training objective and discusses its impact on downstream tasks of semantic segmentation. To validate the pre-training effectiveness, it takes ImageNet as pre-training baseline and three segmentation benchmarks (KITTI, NYU-V2, and Cityscapes) as downstream tasks.

**Strengths:**

1. The idea of pre-training (representation learning) on the geometry task (monocular depth estimation) is novel. The authors prove that the pre-training of depth estimation is beneficial for downstream semantic segmentation.
2. The quantitative results on the given three segmentation benchmarks are valid (KITTI, NYU-V2, Cityscapes), demonstrating the effectiveness of depth estimation as the pre-training objective.

**Weaknesses:**

1. The authors provided many experimental results. However, many key details are missed.
    -  How many training images are used for fine-tuning in the CityScape experiments in Section 4.2? Did the authors use all 20000 images for pre-training? Which backbone models are used for these experiments (ResNet18 or ResNet50)?
    - Similar questions for Section 4.3 for the NYU-V2 dataset.
    - Table 3 (COCO), I assume it reports the results over the testing set of KITTI (rather than COCO)? Also, which dataset do authors use for monocular depth pre-training (KITTI or COCO)?  Did all methods (Depth, MAE, DINO, MOCO v2) use the same pre-training data? Which backbone models are used for these experiments (ResNet18 or ResNet50)? These serve as important references for evaluation and reproduction, which are not clearly specified in this paper.
 2. Where is Table 4.1 (Page 6)?
 3. Potential unfairness in comparison.
    - In Table 2, "Depth" used KITTI as pre-training set while testing on the same dataset. This comparison to the baseline (ImageNet pre-training) is unfair. This is because the training distribution is much closer to the testing distribution for "Depth". In contrast, ImageNet is characterized as object-centric and thus has a pretty large domain gap to KITTI collected by autonomous cars. What about performance comparison on COCO between ImageNet pre-train and Depth pre-train?
    - In Table 3, the "Supervised Segmentation" performance should be results that trained on KITTI instead of MS-COCO if the authors used KITTI as pre-training and testing data.
 4. Lack of analysis on representation transferability. For example, how is the performance of Depth Pre-train over other segmentation datasets such as COCO and ADE20K? How about transferring to object detection? Transferability is one of the major functions of pre-training. The authors should present relevant experimental results.
 5. Despite the authors proving the effectiveness of pre-training with depth estimation, it still requires significant RGB-D data which cannot be collected online for large-scale pre-training data. Compared to self-supervised studies, e.g. MoCo-variants and MAE, this method has limited application scope.

**Questions:**

Please see the weakness.

---

> ### Author Response · Authors · 2023-11-21
> **Author response to the initial review**
>
> Thank you for your thoughtful review.
>
> **W1:** *How many training images are used for fine-tuning in the CityScape experiments in Section 4.2? Did the authors use all 20000 images for pre-training? Which backbone models are used for these experiments?*
>
> **R:** We will make it clear in the revised version. On Cityscapes, we use the default dataset partition for pre-training and finetuning, which includes 20000 binocular image pairs for depth, and 2975 images for semantic fine-tuning (page 8). We use ResNet 50 as the backbone model. On NYU-V2, we also use the dataset’s default settings, which include 407024 images for depth training and 795 images for semantic segmentation fine-tuning.
>
> *Table 3 (COCO), I assume it reports the results over the testing set of KITTI (rather than COCO)?*
>
> **R:** The test set is KITTI. We initialize the backbone with COCO supervised segmentation and verify that monocular depth can close the performance gap between supervised pre-training on the same task and other pre-training tasks.
>
> *Also, which dataset do authors use for monocular depth pre-training (KITTI or COCO)? Did all methods (Depth, MAE, DINO, MOCO v2) use the same pre-training data? Which backbone models are used for these experiments (ResNet18 or ResNet50)? These serve as important references for evaluation and reproduction, which are not clearly specified in this paper.*
>
> **R:** The backbone is Resnet50, as in the captions of Table 3. Depth, optical flow, and MAE are trained in-domain, while dino and MOCO are trained on ImageNet since they are contrastive learning methods that require training on object-centric data. We did reproduce contrastive learning "in-domain" on KITTI but the results are not informative.
>
> **W2:** *Where is Table 4.1 (Page 6)?*
>
> **R:** It’s a typo. We are referring to Table 2. Will correct.
>
> **W3:** *In Table 2, "Depth" used KITTI as pre-training set while testing on the same dataset. This comparison to the baseline (ImageNet pre-training) is unfair. This is because the training distribution is much closer to the testing distribution for "Depth". In contrast, ImageNet is characterized as object-centric and thus has a pretty large domain gap to KITTI collected by autonomous cars. What about performance comparison on COCO between ImageNet pre-train and Depth pre-train?*
>
> **R:** Please see general note for in-domain training. Nonetheless, this is an advantage of pretraining for depth. One does not need to restrict the pretraining data to a particular dataset, but can leverage any of structure-from-motion (video), binocular stereo, or measurements of depth sensor (time-of-flight) to do pretraining.
>
> *In Table 3, the "Supervised Segmentation" performance should be results that trained on KITTI instead of MS-COCO if the authors used KITTI as pre-training and testing data.*
>
> **R:** The reviewer suggests performing supervised pre-training, and apply supervised fine-tuning on the same task and same dataset, which breaks the purpose of pre-training as one can simply used the supervised “pretrained” network for inference without finetuning.
>
> **W4:** *Lack of analysis on representation transferability. For example, how is the performance of Depth Pre-train over other segmentation datasets such as COCO and ADE20K? How about transferring to object detection? Transferability is one of the major functions of pre-training. The authors should present relevant experimental results.*
>
> **R:** Thanks to the reviewer for the suggestion. We are indeed investigating the performance of incorporating depth in pre-training in our follow-up work, especially on different semantic-related tasks. But domain adaptation is beyond the scope of the current paper.
>
> **W5:** *Despite the authors proving the effectiveness of pre-training with depth estimation, it still requires significant RGB-D data which cannot be collected online for large-scale pre-training data. Compared to self-supervised studies, e.g. MoCo-variants and MAE, this method has limited application scope.*
>
> **R:** Monocular depth can be inferred from videos without explicit depth data (i.e., self-supervised), as evidenced in our experiments on KITTI, where we refrain from using any ground-truth depth information. So, given recent advances in camera pose estimation, usage of large-scale online data is viable. However, we note that the compute required to train at that scale is prohibitive for this work, where we aim to conduct numerous controlled experiment to analyze the viability of depth pretraining under different settings.
>
> It is worth mentioning that our intention is not to suggest replacing methods like MoCo or MAE with depth. Instead, we illustrate that depth achieves comparable performance. In real application, it might be advantageous to fine-tune MoCo and MAE using in-domain depth data for optimal results.
>
> We hope this response answers the reviewer's question. If there are further concerns, we are more than willing to provide additional explanations.

---

### Author Response · Authors · 2023-11-16
**A general note to the reviewers and AC:**

We express our sincere appreciation to the reviewers for their valuable feedback. We are delighted that the reviewers collectively agree with our proposal to employ monocular depth estimation, a geometry task, as a pre-training method for semantic segmentation, a semantic task. The consensus on the viability conclusions is encouraging. In response to several critical concerns, we provide clarifications as follows:
___
**Novelty of the Paper:**

We respectfully disagree with the assertion that the paper lacks innovation. We never claim technical novelty simply by using depth. The innovation, articulated in the first paragraph of the introduction, is in answering the following scientific hypothesis:

*Whether pre-training on a GEOMETRY task (monocular depth estimation) can be a viable option for improving a seemingly distantly related SEMANTIC task (semantic segmentation).*

If the answer is “yes” (and it is!), then this opens up possibilities for a new avenue for pre-training that is compatible with how one can cheaply collect data (monocular depth can be inferred from videos without supervision) today.  We are the first to explore this idea in detail and are innovative in meticulously designing experiments that robustly validate it. The manuscript affirms this counterintuitive hypothesis by conducting experiments under 51 settings, among which we highlight examples of noteworthy innovations:

**Not every geometric task is suitable:** We compare with optical flow, another task exhibiting a piecewise-smooth range obtained by minimizing the same photometric error as depth estimation. Despite the similarity, depth aids the task, while flow poses challenges. Detailed analysis is presented on pages 7 and 14-15 in the appendix, elucidating the uniqueness of depth. This has never been demonstrated before; hence, the experiment along with the analysis is novel.

**Addressing object scales in pre-training:** We recognize a crucial but often-overlooked prior, the object scale, in 'object-centric' dataset, and are the first to address a scale mismatch between pre-training and downstream datasets. To validate the effect of the scales, we conducted a cross-resolution experiment (Figure 5 and page 7). Our findings reveal that pre-training on high-resolution data is effective for fine-tuning on low-resolution data, but the reverse is not true.

These novel experimental designs (and more in the paper) are executed with scientific rigor. We firmly believe the paper carries a substantial scientific contribution to the field.
___
**Fairness of using in-domain data for pre-training:**

It is important to note that the goal of this paper is to  *validate the viability of monocular depth as pre-training*, not to rank different methods, although a comparison is necessary since we define viability as achieving on-par or superior performance to the common practice (e.g. ImageNet). Therefore, fairness is represented by the practicality of the proposed hypothesis, not by restricting all training to out-of-domain data. We demonstrate such practicality with three datasets representing different modes for learning monocular depth. Nonetheless, we DO compare with out-of-domain models using DPT (Table 2), and the results align with our conclusion. It is worth mentioning that, our findings are confirmed by recent work [1], which suggests that monocular depth models trained on large-scale data may serve as a general pre-training strategy.

Hence, whether in-domain or out-of-domain, depth pre-training IS A VAIBLE option. However, we note that being able to train ‘in-domain’ is considered an advantage when using depth as pre-training, as discussed in detail in Appendix D.3. Further, although also trained 'in-domain', we show that optical flow, despite being trained by a minimizing a similar reprojection loss, is not a viable option. So the improvement does not simply come from the data but from the task. From an engineering perspective, as stated by Reviewer gFUe: *“It helps the authors conclude multiple guidances and helps convince readers. The conclusions are useful to industry.”*, this paper provides evidence for engineers about the types of data they should consider beyond the current datasets.

That being said, we understand where the reviewers are coming from, and acknowledge the significance of adaptation for different domains. We hope as the importance of depth in representation learning becomes apparent, more large-scale datasets and pre-training will emerge so that a generic depth model becomes possible.

[1] Goldblum, Micah, et al. "Battle of the Backbones: A Large-Scale Comparison of Pretrained Models across Computer Vision Tasks."
___
We appreciate the reviewers' feedback. We will address each concern individually and provide a revised version of the paper at the end of the author-reviewer discussion period. Thanks to the reviewers for their time and valuable input.

---

### Meta-Review · Area_Chair_gkEQ · 2023-12-06

**Metareview:**

The paper receives 1 borderline accept, 1 borderline reject, and 2 reject. The main concerns from the reviewers are: 1) the technical novelty; 2) experimental setting (e.g., dataset, pre-training scheme, evaluation) to validate the proposed method; 3) the applicability of the approach due to the need to the depth data. Although there are some merits in the paper, the AC took a close look at the paper and agrees with reviewers' concerns. Hence, the rejection rating is recommended.

**Justification For Why Not Higher Score:**

There is a concern about the application scope of the proposed method due to the requirement of the depth data for pre-training. Although  there are initial results shown in the paper, it would be more beneficial to conduct more experiments across datasets to verify the claim.

**Justification For Why Not Lower Score:**

N/A

---

### Decision · Program_Chairs · 2024-01-16

Reject